# MRI-Derived Body Composition and Breast Cancer Risk in Postmenopausal Women: UK Biobank Study

**DOI:** 10.3390/cancers17244036

**Published:** 2025-12-18

**Authors:** Livingstone Aduse-Poku, Lusine Yaghjyan, Stephen E. Kimmel, Susmita Datta, Shama D. Karanth, Jae Jeong Yang, Caretia Washington, Dejana Braithwaite

**Affiliations:** 1 Department of Epidemiology, College of Public Health & Health Professions & College of Medicine, University of Florida, 2004 Mowry Road, 2nd Floor, Gainesville, FL 32610, USA; adusepokul@vcu.edu (L.A.-P.); lyaghjyan@ufl.edu (L.Y.); skimmel@ufl.edu (S.E.K.); caretia.washingt@ufl.edu (C.W.); 2Department of Biostatistics, College of Public Health & Health Professions & College of Medicine, University of Florida, Gainesville, FL 32611, USA; susmita.datta@ufl.edu; 3University of Florida Health Cancer Center, University of Florida, Gainesville, FL 32611, USA; shama.karanth@surgery.ufl.edu (S.D.K.); yang.jae@surgery.ufl.edu (J.J.Y.); 4Department of Surgery, College of Medicine, University of Florida, Gainesville, FL 32611, USA

**Keywords:** breast cancer, adipose tissue, skeletal muscle, magnetic resonance imaging, UK biobank

## Abstract

This study examined whether detailed MRI measures of body composition are associated with breast cancer risk in postmenopausal women. Using data from 15,699 participants in the UK Biobank, we found that higher levels of visceral adipose tissue and muscle-fat infiltration were associated with a significantly increased risk of developing breast cancer, independent of traditional risk factors. In contrast, neither subcutaneous fat nor the total muscle volume demonstrated any obvious association. These data suggest that the site and quality of adiposity, instead of overall obesity itself, may contribute to breast cancer development. Incorporating imaging-derived body composition metrics into risk assessment could improve early identification of women at elevated risk and inform targeted prevention strategies.

## 1. Introduction

Breast cancer is the most frequently diagnosed cancer and the primary cause of cancer-related death among women globally, with an incidence of 11.7% [1]. While factors such as hereditary, genetics, and nonhereditary elements contribute to breast cancer occurrence, only 5–10% are attributed to heredity and genetics. This suggests that nonhereditary causes play a major role. Early menarche, alcohol consumption [2,3], late menopause [4], postmenopausal obesity [5], and high levels of endogenous estradiol [6] are all linked with breast cancer as nonhereditary factors. Obesity is particularly noteworthy as one of the few modifiable risk factors, and there is substantial evidence linking it to the development of breast cancer.

According to the Centers for Disease Control and Prevention, approximately 40% of adults in the USA are classified as obese (defined as having a body mass index [BMI] ≥30 kg/m^2^). Scientific evidence suggests that increased adiposity contributes to cancer development through mechanisms such as chronic inflammation, high levels of insulin, elevated leptin, reduced adiponectin levels, abnormal metabolism of sex steroid hormones, and hyperlipidemia [7,8]. Body composition has traditionally been assessed using anthropometric measures, such as BMI, waist circumference (WC), and waist-hip-ratio (WHR). Previous studies have consistently provided evidence that high BMI >25.0 kg/m^2^ is associated with an increased risk of breast cancer in postmenopausal women [9,10,11,12,13]. Similarly, WHR and WC, which are usually used as measures of central obesity, have also been positively associated with breast cancer risk [14,15]. While BMI, WC, and WHR provide general indicators of body fat and health risk, they do not provide detailed information about the distribution of fat and muscle in the body. However, imaging techniques such as Dual-energy X-ray Absorptiometry (DXA), magnetic resonance imaging (MRI), or computed tomography (CT) scans provide a more detailed and accurate picture of body composition. These techniques can differentiate between bone, fat, and muscle mass, and also provide insight into the distribution of fat in different regions of the body. They can precisely distinguish various elements such as fat tissue, muscles, bone, and water. This information is crucial since the location of body fat (visceral vs. subcutaneous) can have different implications for health.

MRI provides three-dimensional data, giving a more complete image of body composition than two dimensional methods like DXA. Additionally, MRI does not utilize ionizing radiation, which can be harmful with repeated exposure. Moreover, MRI provides detailed differentiation of various types of adipose tissues and muscles. This can be very important in studying breast cancer risk as different adipose tissues and skeletal muscles can have different associations with cancer risk. However, most studies used techniques with relatively lower spatial resolution such as CT scans and DXA for body composition assessment. Whole-body fat mass was significantly associated with the risk of breast cancer in two studies conducted using the Women’s Health Initiative (WHI) dataset [16,17]. In one of the largest studies to date (n = 10,931) involving postmenopausal women, the highest category of DXA-derived whole-body fat mass was associated with a two-fold increase in invasive breast cancer risk [17]. Additionally, three studies found a significant link between fat mass of the trunk and breast cancer risk [16,17,18], while one study found no significant association [19].

The study adopted a population-based approach and leveraged high-resolution MRI data to overcome the limitations of the previous studies, which often relied on electronic health records and imaging techniques with lower soft tissue contrast resolution [20,21]. Furthermore, previous studies have only focused on adiposity as an exposure without considering the association between skeletal muscle mass and breast cancer risk. Therefore, this study aims to assess the associations of MRI-assessed adiposity and skeletal muscle volume with breast cancer risk in a population-based cohort.

## 2. Methods

### 2.1. Description of Study Population and Recruitment

This study included postmenopausal women selected from the UK Biobank, a well-established prospective cohort representing around 502,000 adults aged 40–69 years from the general population of the United Kingdom. Upon enrollment, participants provided data on various demographic factors and biological samples (blood, urine, and saliva). The study also includes repeat assessments, web-based questionnaires, multi-modal imaging, and genomic data from stored biological samples. Baseline assessments were conducted between 13 March 2006, and 1 October 2010, across 22 different centers in diverse geographic locations with varied demographics and urban-rural settings. During this time frame, all participants shared detailed health information through questionnaires and interviews. Additionally, they underwent physical examinations, provided blood, urine, and saliva samples, and agreed to be monitored for their health outcomes.

Subsequently, around 103,514 participants residing within approximately 35 km of the UK Biobank Coordinating Centre in Stockport, UK, were invited to participate in the first repeat assessment of the baseline measures. Invitations were extended via email or letter. This first repeat assessment occurred between 1 August 2012, and 7 June 2013, with a completion rate of 20%, totaling 20,346 participants. Furthermore, two more assessments were initiated in 2014 and 2019, respectively, and are ongoing. These assessments are being conducted in a subset of approximately 50,000 participants. Participants’ health outcomes were obtained by connecting their records to National Health Service (NHS) Digital and Public Health England for those in England and Wales and to the NHS Central Register (NHSCR) for those in Scotland. The most recent data from cancer registries, linked to the UK Biobank, was available until 8 December 2023.

The planned study included postmenopausal women enrolled in the UK Biobank who had no prior history of breast cancer at the time of enrollment. Additionally, these women had no other cancer types except for non-melanoma skin cancer when they joined the study. Women were classified as postmenopausal if they met one of the following criteria: (1) reported cessation of periods, (2) bilateral oophorectomy, or (3) hysterectomy with one or both ovaries retained, and were aged 54 or older for those with a history of smoking, or 56 or older for those who had never smoked [22]. UK Biobank study received approval from the North-West Multicenter Research Ethics Committee (MREC), covering the entire UK. All participants in the UK Biobank study provided written informed consent upon recruitment.

### 2.2. MRI-Body Composition Assessment

The study utilized a Siemens MAGNETOM Aera 1.5-T scanner, Forchheim, Germany and employed the dual-echo Dixon Vibe protocol to acquire neck and knee images. Various body composition parameters were evaluated, including abdominal subcutaneous adipose tissue (SAT), visceral adipose tissue (VAT), the volume of thigh muscles, muscle-fat infiltration (MFI) in the front portion of thighs, and liver Proton Density Fat Fraction (PDFF). The image analysis process involved calibrating and stacking the images, segmenting relevant areas, and quantifying fat and muscle volumes [23,24,25]. Manual quality checks were performed by an analysis engineer to ensure accuracy. The analysis was conducted using AMRA Profiler Research software developed by AMRA Medical AB in Linköping, Sweden.

This study included participants from the UK Biobank cohort who have undergone MRI scans to assess their body composition. The registration process used a non-rigid atlas-based method, as described in the reference [23]. VAT refers to adipose tissue inside the abdominal cavity but excludes tissues outside the skeletal muscles in the abdomen, as well as adipose tissue and lipids behind or surrounding the spine and back muscles. SAT refers explicitly to adipose tissue within the subcutaneous area of the abdomen, extending from the top of the femoral head to thoracic vertebrae T9. The posterior thigh muscles include the gluteus, iliacus, adductor, and hamstring muscles, while the anterior thigh muscles consist of the quadriceps femoris and sartorius muscles. Prototypes for the atlases were selected from previously segmented datasets, covering a diverse range of phenotypes in the UK Biobank imaging study. The selection process occurred in two steps: generating histograms of VAT and SAT to identify potential candidates and visually examining many candidates to identify suitable prototypes. All ground truth atlases underwent rigorous inspection and approval by a trained analysis engineer before being utilized in this study.

Quantification of fat and muscle volumes was conducted using a voting scheme that relied on registered labels, as well as intensity-corrected fat and water images. The integration of calibrated fat images with quality-assured labels allowed for the determination of VAT and SAT volumes across all acquisitions. Thigh muscle volumes, including the left anterior, right anterior, left posterior, and right posterior muscles, were determined by calculating the volume of fat-free muscle using the method outlined in Top of Form [23]. Five parameters were analyzed, which include two lean tissue parameters (total fat-free muscle volume and muscle fat infiltration) and three adipose tissue parameters (SAT, VAT, total adipose tissue volume) [26]. We also assessed body composition phenotypes, which were composites based on levels of total adipose tissue volume and total fat-free muscle volume. The cut-offs for the body composition parameters can be found in (Appendix A). The body composition phenotypes consisted of 4 categories such as (i) normal (high muscle/low adiposity), i.e., highest two tertiles of fat-free muscle volume and lowest two tertiles of total adipose tissue volume; (ii) high muscle/high adiposity, i.e., highest tertile of total adipose tissue volume and highest two tertiles of FFMV (iii) low adiposity/low muscle, i.e., lowest two tertiles of total adipose tissue volume and lowest tertile of fat-free muscle volume, and (iv) adverse body composition phenotype, i.e., 3rd tertile of total adipose tissue and 1st tertile of fat-free muscle volume. All body composition analyses were performed in AMRA Profiler^™^ (AMRA AB, Linköping, Sweden).

### 2.3. Covariate Information

The study collected data on risk factors for breast cancer from participant questionnaires, interviews, and physical examinations conducted between 2006 and 2010, as well as in later years (2012–2013, 2014+, and 2019+). The variables to be considered include age, race, index of multiple deprivations (IDM), BMI, height, age at menarche, age at menopause, number of children/age at the birth of the first child, use of postmenopausal hormone therapy, alcohol consumption, smoking habits, level of physical activity, fruit and vegetable intake, surgical history (history of uterus, ovaries, cervical resection), oral contraceptives history, and reproductive history. In the UK Biobank data, only 20% of participants have follow-up data after baseline assessments. However, the data collected at baseline and follow-up are highly correlated.

### 2.4. Outcomes

The endpoints of interest in this study were incident breast cancer (postmenopausal: International Classification of Diseases (ICD)-10 code: C50). In the UK Biobank, cancer diagnoses are determined by linking to national cancer registries in England, Wales, and Scotland. For this study, complete follow-up data were available until 30 January 2024.

### 2.5. Statistical Analyses

First, we summarized the distribution of study characteristics based on body composition phenotypes. The study employed Cox proportional hazards regression to determine the association between MRI-assessed body composition and the risk of developing breast cancer. During the follow-up period, if a woman was diagnosed with a cancer other than breast cancer or deceased, their data were censored at the time of the diagnosis or death, respectively. For women who were cancer-free, the end of follow-up was the date of their last linkage with the Cancer Registry, which was on 30 January 2024. The regression models were developed using three adipose tissue parameters (SAT, VAT, and total adipose tissue volume), two lean tissue parameters (muscle fat infiltration, fat-free muscle volume), and body composition phenotypes. The body composition measures were analyzed following categorization by tertiles. The time to diagnosis breast cancer was the underlying timescale. Participants were monitored from when they joined the study until they were diagnosed with breast cancer, withdrew from the study, died, or until the study ended. If participants did not experience the specific event of interest, die, or leave the study before the end of follow-up, they were censored. All models were adjusted for age at recruitment (years; continuous), socioeconomic status (based on Townsend deprivation index categorized by quintiles), race (White, Other), alcohol consumption (never, former, current), smoking status (never, former, current), physical activity (MET-min/week continuous), hormone replacement therapy (no, yes), bilateral oophorectomy (no, yes), diabetes (no, yes), and hypertension (no, yes). We assessed the violation of the proportional hazards assumption by conducting tests using Schoenfeld residuals; no violation was found (Appendix A) To account for the potential issue of reverse causation, we performed sensitivity analyses by excluding participants with a follow-up time of two years or less. To evaluate potential nonlinear associations between VAT and MFI with breast cancer risk and to capture variations across the full spectrum of these relationships, we employed restricted cubic splines with 3 knots.

Subgroup analyses were conducted based on BMI (<25 kg/m^2^ versus ≥25 kg/m^2^), waist circumference (<80 cm versus ≥80 cm), and waist-hip ratio (<0.80 and ≥0.80). Interaction terms of these factors with VAT and MFI were generated and added to the model. A Wald test was used to assess whether the interaction was significant. All statistical analyses were performed using SAS software, Version 9.4 (SAS Institute Inc., Cary, NC, USA, 2017) and R statistical software (version 4.0.2). All *p*-values were two-sided.

## 3. Results

The final study sample comprises 15,699 postmenopausal women with complete MRI body composition data, among whom 918 were diagnosed with breast cancer during the follow-up perio (Figure 1). The median follow-up time from baseline to cancer diagnosis was 6.70 years for women diagnosed with breast cancer and 9.68 years for those without breast cancer. The mean age of the participants was 58.58 (SD = 5.18) years (Table 1). The average age of menarche and menopause was 12.63 (SD = 2.57) years and 47.29 (SD = 12.24) years, respectively. Participants with adverse body composition were older than those with normal body composition, with a mean age = 59.10 (5.13) and 57.41 (5.18), respectively. The mean IDM for participants with the adverse body composition phenotype was 16.89 (SD = 13.21), higher than that for participants with a normal body composition phenotype, 13.92 (SD = 10.94). The participants had an average BMI of 26.07 (SD = 3.87) kg/m^2^. The mean summed MET minutes per week was higher among participants with the normal body composition phenotype versus those with adverse body composition, at 3056.31 (2715.68) minutes and 2171.27 (2124.07) minutes, respectively. The majority of the study participants were White, accounting for 14,296 (91.4%). The proportion of participants who had taken hormone replacement therapy and had undergone bilateral oophorectomy was 45% (n = 7041) and 4.1% (n = 646), respectively.

In the age-adjusted Cox proportional hazard models (Table 2), high VAT and total adipose tissue (TAT) were associated with an increased risk of breast cancer among postmenopausal women (3rd vs. 1st tertile HR = 1.26, 95% CI, 1.10–1.47, model 2) and (3rd vs. 1st tertile HR = 1.18, 95% CI, 1.01–1.38, model 2), respectively. In the multivariable Cox model, high VAT was associated with an increased breast cancer risk (3rd vs. 1st tertile aHR = 1.24, 95% CI, 1.10–1.45, model 3). In age and multivariable Cox models, SAT and FFMV were not significantly associated with breast cancer risk. High levels of MFI were associated with an increased risk of breast cancer in the age-adjusted model (3rd vs. 1st tertile HR = 1.54, 95% CI, 1.26–1.89 model 2) and in the multivariable Cox proportional hazard regression model (3rd vs. 1st tertile aHR = 1.53, 95% CI, 1.25–1.87, model 3). In the age-adjusted model, adverse body composition was associated with an increased risk of breast cancer compared to the normal body composition phenotype, HR of 1.24 (95% CI, 1.02–1.55, model 2). However, in the multivariable-adjusted model, body composition phenotype was not significantly associated with breast cancer risk, with an aHR of 1.23 (95% CI, 0.98–1.54, model 3). To reduce the risk of reverse causality, we performed a sensitivity analysis by excluding women with breast cancer diagnosed within 2 years of follow-up (Table 3). High levels of VAT were associated with an increased risk of breast cancer in the age-adjusted model (3rd vs. 1st tertile HR = 1.22, 95% CI, 1.02–1.44, model 2) and the multivariable Cox proportional hazard regression models (3rd vs. 1st tertile aHR = 1.24, 95% CI, 1.05–1.48, model 3). Similarly, high levels of MFI were associated with an increased risk of breast cancer in the age-adjusted model (3rd vs. 1st tertile aHR = 1.33, 95% CI, 1.07–1.65, model 2) and in the multivariable Cox proportional hazard regression model (3rd vs. 1st tertile aHR = 1.35, 95% CI, 1.08–1.68, model 3). There was no association between body composition phenotypes and breast cancer risk, with an aHR of 1.20 (95% CI, 0.94–1.52, model 3). There was no association between body composition phenotypes and breast cancer risk, with an aHR of 1.20 (95% CI, 0.94–1.52, model 3). The HR for VAT volume was close to 1.0 at lower volumes and then started to increase progressively from ~3–4 L, thus clearly showing a J-shaped curve. At larger volumes (>7 L), the slope increased, in accordance with an exponential-like rise in breast cancer risk, Figure 2.

For muscle-fat infiltration, the association showed a milder yet nonlinear pattern, with minimal change in hazard ratio below ~9 L and a more accelerated increase at higher volumes, Figure 3.

In subgroup analyses, we found that VAT significantly interacted with WHR in relation to breast cancer risk. Specifically, the effect of high VAT vs. low VAT on breast cancer was stronger in postmenopausal women with WHR < 0.80, (WHR < 0.80: aHR _[__High VAT vs. low VAT]_, 1.31; 95% CI, 1.02–1.70; WHR ≥ 0.80: aHR _[High VAT vs. low VAT]_, 0.91; 95% CI, 0.67–1.21; p-interaction = 0.025) (Table 4). Additionally, we observed that MFI significantly interacted with BMI in relation to breast cancer risk. Specifically, the effect of high MFI vs. low MFI on breast cancer was stronger in postmenopausal women with BMI ≥ 25 kg/m^2^ (BMI ≥ 25: aHR _[High MFI vs. low MFI]_, 1.88; 95% CI, 1.37–2.59; BMI < 25: aHR _[High MFI vs. low MFI]_, 1.69; 95% CI, 1.23–2.32; p-interaction = 0.025) (Table 5). Lastly, we observed that MFI significantly interacted with waist circumference and WHR in relation to breast cancer risk.

For some factors such as age at recruitment, age at menarche, age at menopause, and index of multiple deprivation, which are of interest in relation to breast cancer, there was no statistical difference (*p* > 0.05) between participants who had at least two scans compared to had one scan and those who did not take MRI scans (Appendix A). For factors such as race, smoking status, alcohol drinker status, HRT, and bilateral oophorectomy, there is a statistically significant difference in all variables between participants who had at least two scans compared to those who had one scan and those who did not take MRI scans.

## 4. Discussion

This is one of the first large-scale study to assess the association between MRI-assessed adiposity and skeletal muscle volume with breast cancer risk. In this study, we utilized a population-based sample of 15,669 women with MRI-assessed body composition data. The average age of the participants was 58.6 years, and they were followed for an average of 6.70 years. In the primary multivariable Cox proportional hazard model, high VAT and MFI showed an association with an increased risk of breast cancer. Participants with higher VAT and MFI had a 24% and 53% higher risk of breast cancer, respectively. These associations remained consistent even after conducting a sensitivity analysis, in which women diagnosed with breast cancer within two years of follow-up were excluded. Additionally, we observed a J-shaped positive association between VAT and MFI in relation to breast cancer risk. Furthermore, we found that high MFI was associated with an increased risk of breast cancer in women with a BMI of 25 kg/m^2^ or higher but not in women with a BMI below 25 kg/m^2^. Moreover, high MFI was found to be associated with an increased risk of breast cancer in women with a waist circumference of 80 cm or more and a WHR of 0.80 or more.

Adipose tissues are typically made up of subcutaneous and visceral fat. SAT is the layer of fat under the skin, while visceral adipose tissue is found within the abdominal cavities, also known as organ fat or intra-abdominal fat. In this study, we found that a high level of VAT was associated with an increased risk of breast cancer among postmenopausal women. However, no significant association was observed between high levels of SAT and breast cancer risk. This observation is not surprising since VAT is more metabolically active and responsive to hormonal fluctuations compared to SAT [27,28]. Also, we observed an exponential-shaped positive association between VAT and breast cancer risk. This suggests a threshold effect or a point (VAT = 3.00 L) at which the risk of breast cancer begins to increase rapidly with greater amount of VAT. This finding suggests that there may be a threshold level of visceral adipose tissue below which the risk does not significantly increase but beyond which the risk of adverse events rises sharply. This could be vital for setting clinical targets or guidelines for patient health. Studies on the association between specific adipose tissues (VAT and SAT) and breast cancer risk are limited [29]. Our findings are consistent with those of previous prospective studies that used imaging techniques for body composition assessment. In previous studies, total fat mass and trunk fat mass were used using imaging techniques such as CT and DXA. DXA-derived whole-body fat mass was significantly associated with the risk of breast cancer in two studies conducted using the Women’s Health Initiative (WHI) dataset [16,17]. In one of the largest studies to date (n = 10,931), which included postmenopausal women, the highest versus lowest category of DXA-derived body whole-body fat mass was found to be associated with a doubling of the risk of invasive breast cancer (HR: 2.17; 95% CI: 1.54–3.05) [17]. Three of the studies found a significant association between trunk fat mass and breast cancer risk [16,17,18]. In contrast, one study found no significant association between trunk fat mass and breast cancer risk [19]. In a prospective study of 445 participants, Cao and colleagues found no significant association between CT-assessed VAT and breast cancer risk. To better contextualize our findings, we also compared them with studies conducted in other global populations. Research from Asian cohorts, including large Korean and Chinese studies, has similarly demonstrated that visceral adiposity and central fat distribution are associated with elevated breast cancer risk, even among women with normal BMI [30,31,32]. These observations align with our results and suggest that the biological impact of VAT and MFI on breast cancer risk may transcend geographic and ethnic differences.

Menopause affects body composition by altering the production of sex steroid hormones. As women enter this phase, there is a gradual decrease in estrogen production in the ovaries. The endocrine function is then taken over by converting androstenedione in the peripheral adipose tissues located on the periphery of the adipose tissues [33]. This change results in lower estrogen levels in the bloodstream. Combined with the common shift towards a more sedentary lifestyle after menopause, this can lead to lower cholesterol levels and potentially an increase in visceral adipose tissue (VAT) [33]. The excess VAT causes fat to accumulate in the liver, providing free fatty acids through the portal vein. As fat accumulates in the liver, its function deteriorates, leading to decreased levels of sex hormone-binding globulin (SHBG). This decline in SHBG contributes to higher levels of free estrogen in the bloodstream [34], which is associated with an increased risk of breast cancer among postmenopausal women [35]. Another proposed mechanism is through obesity-associated insulin resistance. Insulin is a hormone that regulates glucose and lipid homeostasis. Dysregulation of insulin signaling can activate the PI3K/Akt/mTOR and Ras/Raf/MAPK pathways, which may enhance cell proliferation and increase the risk of neoplasia in the breast [36,37]. Insulin also acts as a growth factor, and insulin-like peptides (ILPs), such as IGF-1 and IGF-2, can bind to the insulin receptor and trigger intracellular responses similar to those triggered by insulin [38]. Elevated levels of insulin and IGF-1 in obesity create an environment that promotes cell growth and inhibits cell death, accelerating the accumulation of mutations and favoring carcinogenesis [35]. Another proposed mechanism is the hypoxia-angiogenesis process. As adipose tissue expands, hypoxia develops due to reduced oxygen concentrations, leading to an obesity-associated inflammatory response [39]. The hypoxic and angiogenic environment of obese adipose tissue, along with elevated levels of cytokines in obese individuals, may promote cancer development [40]. Together, these endocrine, metabolic, and inflammatory pathways illustrate multiple biologically plausible mechanisms through which menopause-related changes in body composition, particularly increases in VAT and alterations in muscle and adipose tissue quality, may contribute to elevated breast cancer risk (Appendix A).

One distinguishing feature of MRI is its ability to provide a detailed, high-resolution 3D view of all body tissues, including adipose tissue (fat), muscle tissue, bone, and the amount of fat within the muscles known as MFI [41]. MFI, also known as myosteatosis, is the pathologic accumulation of fat in skeletal muscle resulting in poor metabolic and musculoskeletal health. MFI is now recognized as a separate condition from sarcopenia. Sarcopenia is commonly associated with aging and involves the decline of muscle strength, architecture, contraction, and capacity [42,43]. Fat deposition within skeletal muscles occurs in two primary mechanisms: intermuscular adipose tissue, which accumulates between muscle fibers (cells) and between the outer layers of skeletal muscles, and intramuscular fat, which is found within the inner layers. Additionally, lipid droplets can accumulate within muscle fibers themselves, known as intramyocellular lipid deposits [42]. Numerous factors such as aging, illness, and muscle damage have been identified in prior research as potential contributors to fat accumulation within skeletal muscle [44,45]. This process is influenced by a range of regulators, genes, and signaling pathways. The impact of muscle-fat infiltration on breast cancer has not been well-studied. In this study, we found that a high level of MFI was associated with an increased risk of breast cancer in postmenopausal women. Similarly, in a case–controlled study, high versus low intramuscular adipose tissue was associated with an increased risk of TNBC subtype [46].

Anthropometric measures, such as BMI, waist circumference, and WHR, are commonly used to assess adiposity. However, the reliability of BMI in predicting the risk of disease among individuals with excess adiposity is questionable due to its limitations in distinguishing between lean and fat mass. Despite this, epidemiological studies assessing the association between anthropometrically measured body composition and breast cancer prognosis have found conflicting results. In some studies, higher BMI [30,31], weight gain [32], waist circumference [13], and hip circumference [13] were associated with an increased risk of breast cance [32]. In this study, the effect of high MFI on breast cancer was stronger in postmenopausal women with BMI ≥ 25 kg/m^2^, waist circumference ≥ 80 cm, and WHR ≥ 0.80 compared to their counterparts. These findings underscore the importance of considering body composition, alongside traditional measures of obesity, in breast cancer risk assessments and potentially tailoring prevention strategies for those at higher risk due to these factors. This suggests that body composition metrics assessed by MRI could be valuable biomarkers for identifying individuals at a higher risk of breast cancer, especially among women with elevated levels of proxy adiposity measures.

We used Schoenfeld residuals to test the proportional hazards assumption concerning various body composition parameters during our model fit assessment. The assessment showed no meaningful breach of the proportional hazards assumption across all variables since *p*-values spanned between 0.19 and 0.58. The analysis demonstrates that MRI-derived body composition parameters do not introduce time-dependent biases in Cox regression estimates. The results demonstrate that hazard ratios remain constant over time which verifies our model’s robustness.

To our knowledge, this study is one of the first to examine the association of MRI-assessed adipose tissues and skeletal muscles with breast cancer risk in postmenopausal patients. It utilized a prospective design, a large sample size, and long-term follow-up, employing standardized procedures administered by trained personnel to measure anthropometric variables and body composition utilizing MRI. Furthermore, this study utilized population-based data compared to prior studies relying on electronic health records. The use of population-based data helps to ensure that the sample is representative of the general population and minimizes selection bias, thereby strengthening the validity and robustness of the findings. In breast cancer patients, MRI is the most commonly used imaging technique, reducing the risk of selection bias and accurately distinguishing between different types of adipose tissues. However, there may be risk of selection bias as participants who had at least two scans compared to those who had one scan and those who did not take MRI scans were significantly different in relation to factors such as race, smoking status, alcohol drinker status, hormone replacement therapy, and bilateral oophorectomy. Also, due to insufficient histological information, it was not feasible to further analyze the impact of fat mass in each type of breast cancer. Residual confounding remains possible despite adjustment for multiple established risk factors. We did not have detailed information on diet quality or dietary patterns, longitudinal weight change, statin or metformin use, or direct measures of physical fitness beyond self-reported METs.

In addition, as the cohort primarily consisted of Europeans, the findings could be limited to this specific population. Also, since this is an observational study, there may be concerns regarding unmeasured confounders. Additionally, this research only utilized baseline assessment data on covariates. Nonetheless, given the strong correlations with values from follow-up assessments, using baseline risk factor data is unlikely to result in information bias [47]. Additionally, there may be a time lag, resulting in a delay between the diagnosis of breast cancer and its inclusion in the national cancer registry, which could lead to outdated or incomplete data.

## 5. Conclusions

Our study provides valuable insights into the relationship between MRI-assessed adiposity and skeletal muscle volume in relation to breast cancer risk, especially in a large-scale population-based cohort. The findings emphasize the association of high VAT and MFI with increased risk of breast cancer, highlighting the potential importance of using these imaging-derived metrics to assess risk and implement preventive strategies. Further research is necessary to better understand the underlying mechanisms linking VAT and MFI to breast cancer risk. Additionally, it is important to investigate whether interventions aimed at reducing fat mass, such as diet and exercise programs or medications like aromatase inhibitors, can effectively lower the risk of breast cancer in postmenopausal women.

## Figures and Tables

**Figure 1 cancers-17-04036-f001:**
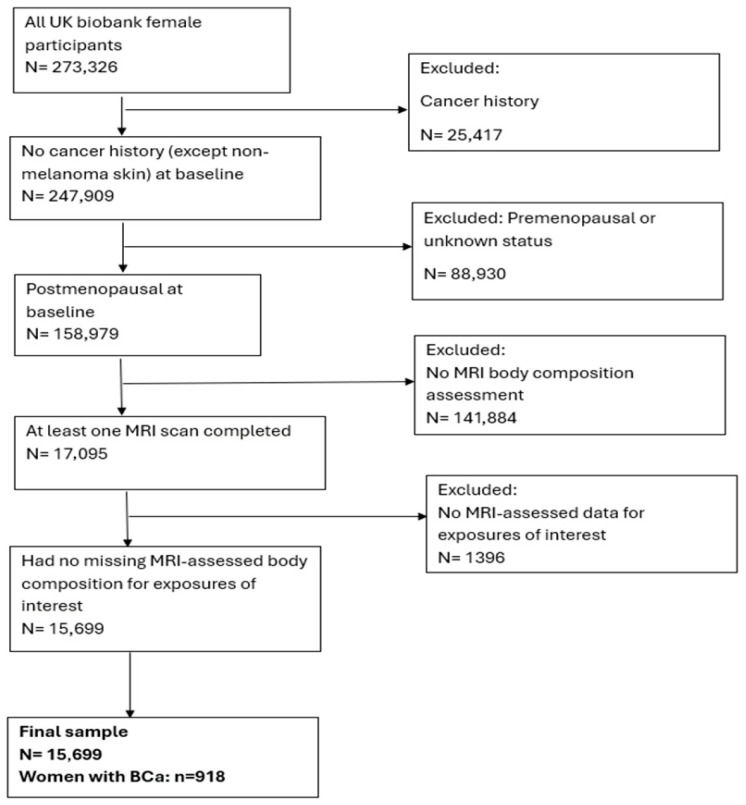
Study Sample Selection Diagram.

**Figure 2 cancers-17-04036-f002:**
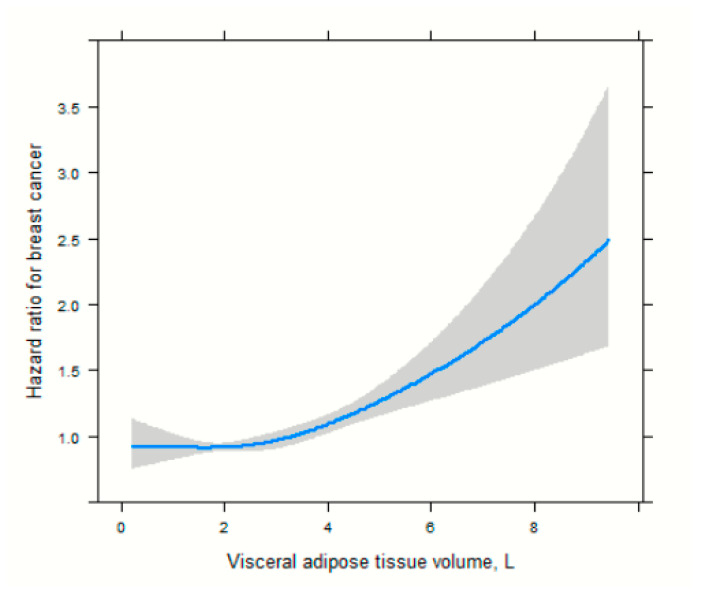
Spline functions with corresponding 95% CIs from Cox proportional hazards regression for the relations of visceral adipose tissue.

**Figure 3 cancers-17-04036-f003:**
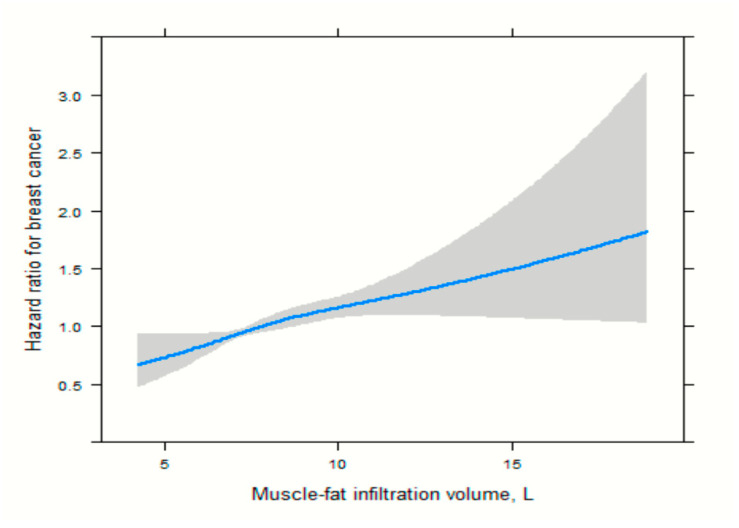
Spline functions with corresponding 95% CIs from Cox proportional hazards regression for the relations of muscle-fat infiltration.

**Table 1 cancers-17-04036-t001:** Characteristics of study participants.

Characteristics	Overall	Normal	High Adiposity/High Muscle	Low Adiposity/Low Muscle	Adverse Body Composition Phenotype (High Adiposity/Low Muscle)
N	15,669	2517	2247	8289	2616
Mean (SD)					
Age at recruitment, years	58.58 (5.18)	57.41 (5.18)	57.41 (5.12)	59.08 (5.10)	59.10 (5.13)
Age at menarche, years	12.63 (2.57)	12.78 (2.59)	12.54 (2.56)	12.66 (2.57)	12.46 (2.57)
Age of menopause, years	47.29 (12.24)	48.05 (10.94)	47.88 (11.03)	47.06 (12.75)	46.80 (12.72)
Index of multiple Deprivation	15.01 (11.69)	13.92 (10.74)	16.16 (11.94)	14.41 (11.28)	16.98 (13.21)
Body mass index (BMI)	26.07 (3.87)	24.93 (2.76)	31.05 (4.57)	23.96 (2.85)	29.61 (3.87)
Waist circumference (cm)	82.43 (10.77)	79.85 (7.20)	77.17 (7.70)	94.66 (10.58)	89.61 (9.08)
Hip circumference	101.71 (8.95)	100.70 (5.94)	111.49 (9.18)	97.42 (6.27)	107.86 (8.36)
Waist-hip ratio (WHR)	1.25 (0.10)	1.26 (0.09)	1.28 (0.09)	1.18 (0.09)	1.20 (0.09)
Summed MET minutes per week	2662.04 (2377.84)	3056.31 (2715.68)	2766.05 (2323.21)	2408.40 (2316.24)	2171.27 (2124.07)
N (%)					
Race White Other races	14,296 (91.4) 1340 (8.6)	2288 (91.2) 222 (8.8)	2054 (91.7) 185 (8.3)	7564 (91.4) 709 (8.6)	2390 (91.4) 224 (8.6)
Smoking status Never Former Current	9612 (61.5) 5296 (33.9) 724 (4.6)	1567 (62.4) 844 (33.6) 99 (3.9)	1297 (57.9) 814 (36.3) 131 (5.8)	5234 (63.2) 2681 (32.4) 361 (4.4)	1514 (58.1) 957 (36.8) 133 (5.1)
Alcohol drinker status Never Former Current	515 (3.3) 345 (2.2) 14,802 (94.5)	52 (2.1) 40 (1.6) 2423 (96.3)	74 (3.3) 56 (2.5) 2115 (94.2)	271 (3.3) 174 (2.1) 7842 (94.6)	118 (4.5) 75 (2.9) 2422 (92.6)
Fruit intake No Yes	273 (11.3) 2140 (88.7	41 (9.5) 391 (90.5)	53 (14.9) 302 (85.1)	119 (9.3) 1159 (90.7)	60 (17.2) 288 (82.8)
Vegetable intake No Yes	299 (12.4) 2114 (87.6)	53 (12.3) 379 (87.7)	44 (12.4) 311 (87.6)	149 (11.7) 1129 (88.3)	53 (15.2) 295 (84.8)
Hormone replacement therapy No Yes	8611 (55.0) 7041 (45.0)	1547 (61.5) 968 (38.5)	1306 (58.2) 938 (41.8)	4450 (53.8) 3829 (46.2)	1308 (50.0) 1306 (50.0)
Bilateral oophorectomy No Yes	14,946 (95.9) 646 (4.1)	2427 (96.9) 79 (3.1)	7937 (96.2) 312 (3.8)	2129 (95.1) 110 (4.9)	2443 (94.4) 145 (5.6)
Diabetes No Yes	15,359 (98.1) 299 (1.9)	2487 (98.9) 28 (1.1)	8193 (98.9) 93 (1.1)	2168 (96.6) 77 (3.4)	2511 (96.1) 101 (3.9)
Hypertension No Yes	10,683 (68.2) 4986 (31.18)	1930 (76.7) 587 (23.3)	6046 (72.9) 2243 (27.1)	1273 (56.7) 974 (43.3)	1434 (54.8) 1182 (45.2)

**Table 2 cancers-17-04036-t002:** Associations of measures of adiposity and skeletal muscles with breast cancer risk among postmenopausal women.

Body Composition Measures	Cases/Person-Years	Unadjusted HR (95% CI) Model 1	Age-Adjusted (95% CI) Model 2	aHR (95% CI) Model 3
Visceral adipose tissue (VAT) Low Medium High	277/2286.03 291/2213.35 350/2815.24	Ref 1.04 (0.88–1.23) 1.27 (1.09–1.49)	Ref 1.03 (0.88–1.22) 1.26 (1.10–1.47)	Ref 1.04 (0.88–1.22) 1.24 (1.10–1.45)
Subcutaneous adipose tissue (SAT) Low Medium High	298/2469.67 293/2362.22 327/2482.72	Ref 0.97 (0.83–1.14) 1.09 (0.93–1.27)	Ref 0.97 (0.82–1.14) 1.10 (0.94–1.28)	Ref 0.96 (0.81–1.13) 1.08 (0.92–1.26)
Total adipose tissue (TAT) Low Medium High	310/2567.22 290/2265.15 318.2482.24	Ref 0.99 (0.84–1.16) 1.17 (1.01–1.37)	Ref 0.98 (0.84–1.16) 1.18 (1.01–1.38)	Ref 0.98 (0.81–1.15) 1.17 (1.01–1.37)
Fat-free muscle volume Low Medium High	221/1755.41 205/1492.76 275/2343.76	Ref 0.93 (0,77–1.13) 0.98 (0.81–1.19)	Ref 0.95 (0.78–1.15) 1.02 (0.85–1.24)	Ref 0.94 (0.78–1.14) 1.02 (0.84–1.23)
Muscle fat infiltration Low Medium High	163/1228.47 223/1714.05 276/2344.79	Ref 1.38 (1.12–1.68) 1.61 (1.32–2.00)	Ref 1.34 (1.09–1.64) 1.54 (1.26–1.89)	Ref 1.34 (1.09–1.64) 1.53 (1.25–1.87)
Body composition phenotypes Normal Low adiposity/low muscle High adiposity/high muscle Adverse body composition	135/1097.00 465/3735.38 138/1158.84 180/1323.41	Ref 1.15 (0.91–1.46) 1.05 (0.87–1.27) 1.29 (1.03–1.61)	Ref 1.01 (0.84–1.23) 1.15 (0.90–1.45) 1.24 (1.02–1.55)	Ref 1.02 (0.854–1.24) 1.14 (0.90–1.44) 1.23 (0.98–1.54)

aHR, adjusted hazard ratio, CI, confidence interval, statistically significant. Model 2 adjusted for age. Model 3 adjusted for age at recruitment (years; continuous), socioeconomic status (based on Townsend deprivation index categorized by quintiles), ethnicity, alcohol drinker status, smoking status, fruit intake, vegetable intake, physical activity, hormone replacement therapy, bilateral oophorectomy, diabetes, and hypertension.

**Table 3 cancers-17-04036-t003:** Sensitivity analyses: Associations of measures of adiposity and skeletal muscles with breast cancer risk among postmenopausal women, excluding women with breast cancer diagnosed within 2 years of follow-up.

Body Composition Measures	Cases/Person-Years	Unadjusted HR (95% CI) Model 1	Age-Adjusted (95% CI) Model 2	aHR (95% CI) Model 3
Visceral adipose tissue (VAT) Low Medium High	237/2242.72 257/2179.19 300/2767.74	Ref 1.13 (0.95–1.35) 1.22 (1.03–1.45)	Ref 1.13 (0.95–1.35) 1.22 (1.02–1.44)	Ref 1.15 (0.96–1.37) 1.24 (1.05–1.48)
Subcutaneous adipose tissue (SAT) Low Medium High	269/2437.29 248/2319.75 277/2432.61	Ref 0.94 (0.79–1.11) 1.06 (0.90–1.26)	Ref 0.94 (0.79–1.11) 1.08 (0.91–1.28)	Ref 0.94 (0.79–1.11) 1.10 (0.93–1.30)
Total adipose tissue (TAT) Low Medium High	272/2526.59 253/2228.36 269/2434.70	Ref 0.99 (0.83–1.17) 1.12 (0.95–1.33)	Ref 0.99 (0.83–1.17) 1.13 (0.96–1.34)	Ref 0.99 (0.83–1.17) 1.15 (0.97–1.37)
Fat-free muscle volume Low Medium High	184/1720.81 174/1461.98 188/1693.44	Ref 0.96 (0.78–1.19) 1.02 (0.83–1.25)	Ref 0.98 (0.80–1.21) 1.06 (0.86–1.30)	Ref 0.96 (0.78–1.19) 1.04 (0.85–1.28)
Muscle fat infiltration Low Medium High	143/1207.26 192/1679.80 211/1989.18	Ref 1.35 (1.09–1.68) 1.39 (1.12–1.72)	Ref 1.32 (1.06–1.64) 1.33 (1.07–1.65)	Ref 1.33 (1.06–1.65) 1.35 (1.08–1.68)
Body composition phenotypes Normal Low adiposity/low muscle High adiposity/high muscle Adverse body composition	121/1081.23 404/3673.72 120/1144.43 149/1290.27	Ref 1.02 (0.83–1.25) 1.08 (0.84–1.40) 1.19 (0.94–1.52)	Ref 0.99 (0.80–1.21) 1.09 (0.85–1.41) 1.16 (0.91–1.48)	Ref 1.00 (0.81–1.22) 1.12 (0.87–1.44) 1.20 (0.94–1.52)

aHR, adjusted hazard ratio, CI, confidence interval, statistically significant. Model 2 adjusted for age. Model 3 adjusted for age at recruitment (years; continuous), socioeconomic status (based on Townsend deprivation index categorized by quintiles), ethnicity, alcohol drinker status, smoking status, fruit intake, vegetable intake, physical activity, hormone replacement therapy, bilateral oophorectomy, diabetes, and hypertension.

**Table 4 cancers-17-04036-t004:** Association between measures of visceral adipose tissue and breast cancer risk in subgroups of anthropometric measures.

Anthropometric Measures	HR (95% CI)	p-Interaction
BMI (<25), kg/m^2^ Low VAT Medium VAT High VAT	Ref 1.12 (0.90–1.40) 1.22 (0.89–1.67)	0.187
BMI (≥25), kg/m^2^ Low VAT Medium VAT High VAT	Ref 0.88 (0.66–1.17) 0.99 (0.76–1.29)	0.863
Waist circumference (<80), cm Low VAT Medium VAT High VAT	Ref 1.13 (0.90–1.41) 1.03 (0.70–1.52)	0.504
Waist circumference (≥80), cm Low VAT Medium VAT High VAT	Ref 0.87 (0.66–1.14) 0.96 (0.75–1.23	0.273
Waist-hip ratio (<0.80) Low VAT Medium VAT High VAT	Ref 0.96 (0.72–1.28) 1.27 (1.39–1.65)	0.002
Waist-hip ratio (≥0.80) Low VAT Medium VAT High VAT	Ref 1.20 (0.98–1.48) 0.88 (0.66–1.18)	0.066

Abbreviations; HR: Hazard ratio, CI: Confidence interval, BMI: Body mass index, WHR: Waist-hip ratio, VAT: Visceral adipose tissue.

**Table 5 cancers-17-04036-t005:** Association between measures of muscle-fat infiltration and breast cancer risk in subgroups of anthropometric measures.

Anthropometric Measures	HR (95% CI)	p-Interaction
BMI (<25), kg/m^2^ Low MFI Medium MFI High MFI	Ref 1.47 (1.13–1.90) 1.21 (0.88–1.68)	0.010
BMI (≥25), kg/m^2^ Low MFI Medium MFI High MFI	Ref 1.35 (0.96–1.90) 1.69 (1.23–2.32)	0.047
Waist circumference (<80), cm Low MFI Medium MFI High MFI	Ref 1.39 (1.06–1.82) 1.10 (0.77–1.56)	0.050
Waist circumference (≥80), cm Low MFI Medium MFI High MFI	Ref 1.36 (0.99–1.86) 1.61 (1.21–2.16)	0.038
Waist-hip ratio (<0.80) Low MFI Medium MFI High MFI	Ref 1.39 (1.02–1.90) 1.55 (1.16–2.08)	0.012
Waist-hip ratio (≥0.80) Low MFI Medium MFI High MFI	Ref 1.36 (1.04–1.77) 1.34 (1.00–1.79)	0.054

Abbreviations; HR: Hazard ratio, CI: Confidence interval, BMI: Body mass index, WHR: Waist-hip. ratio.

## Data Availability

The datasets presented in this article are not readily available because the data are only available from the UK Biobank according to standard controlled access procedures. Requests to access the datasets should be directed to https://www.ukbiobank.ac.uk/use-our-data/apply-for-access/ (accessed on 4 December 2023).

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
