# Peer review of "MRI-Derived Body Composition and Breast Cancer Risk in Postmenopausal Women: UK Biobank Study"

_cancers, 2025, doi:10.3390/cancers17244036_

Round 1

Reviewer 1 Report

Comments and Suggestions for Authors

This is a well designed observational study in a large dataset of women.  The strengths of this study include the sample size, the use of multiple point analysis, and the clinical data that accompany the MRI body composition data.  The following suggestions are included in an attempt to strengthen the manuscript:

1) Although this is an observational study, there is no hypothesis stated.  Did the Authors have a hypothesis that directed the analysis?

2) The numerous statistical associations are well done.  However, I was surprised that the actual data for skeletal muscle and adipose volumes are not presented, even in the Supplementary Data.  The Methods describes the analysis of numerous skeletal muscles including muscles on the anterior and posterior of the thigh.  These values should be presented, to allow readers or other researchers to evaluate.  Also, the percentage of fat infiltration into skeletal muscle is not available, despite this being a significant association for breast cancer risk.  This is my biggest fault of the manuscript, the lack of data for muscle and adipose tissue volumes that resulted in the significant associations with breast cancer risk.  If the Authors suggest that MRI-derived values for muscle and fat should be considered to assess breast cancer risk, than these values should be presented.    

3) Minor - the paragraph starting on line 367 in the Discussion - This paragraph lists a few different proposed mechanisms but there is not conclusion as to what the mechanisms are associated with or proposed to do.  The Authors write "Another proposed mechanism" but there is no clarification on what this means.  As written, this paragraph does not add much to the paper and should be revised to provide clarify on the Authors intent.  

Author Response

Comment #1: This is a well designed observational study in a large dataset of women.  The strengths of this study include the sample size, the use of multiple point analysis, and the clinical data that accompany the MRI body composition data.  The following suggestions are included in an attempt to strengthen the manuscript:

1) Although this is an observational study, there is no hypothesis stated.  Did the Authors have a hypothesis that directed the analysis?

2) The numerous statistical associations are well done.  However, I was surprised that the actual data for skeletal muscle and adipose volumes are not presented, even in the Supplementary Data.  The Methods describes the analysis of numerous skeletal muscles including muscles on the anterior and posterior of the thigh.  These values should be presented, to allow readers or other researchers to evaluate.  Also, the percentage of fat infiltration into skeletal muscle is not available, despite this being a significant association for breast cancer risk.  This is my biggest fault of the manuscript, the lack of data for muscle and adipose tissue volumes that resulted in the significant associations with breast cancer risk.  If the Authors suggest that MRI-derived values for muscle and fat should be considered to assess breast cancer risk, than these values should be presented.   

Response #1: Thank you for this thoughtful and constructive comment. We agree that providing the underlying MRI-derived skeletal muscle and adipose tissue measurements would enhance transparency and allow readers to more fully evaluate the results. In response, we have now included a comprehensive table of summary statistics for all muscle and adipose tissue volumes analyzed in the study, including fat-free muscle volume, total adipose tissue volume, subcutaneous adipose tissue volume, visceral adipose tissue volume, and muscle fat infiltration. These data are now presented in the Supplementary Materials (Table 4).

Comment #2: Minor - the paragraph starting on line 367 in the Discussion - This paragraph lists a few different proposed mechanisms but there is not conclusion as to what the mechanisms are associated with or proposed to do.  The Authors write "Another proposed mechanism" but there is no clarification on what this means.  As written, this paragraph does not add much to the paper and should be revised to provide clarify on the Authors intent.  

Response #2: Thank you for this helpful comment. The revised paragraph provides a more coherent description of the hormonal, metabolic, and inflammatory pathways involved, and clarifies the intent and conclusions of this section (Page 13, Line 416-420).

Reviewer 2 Report

Comments and Suggestions for Authors

Obesity is a major global health concern, with strong evidence linking excess body weight to several cancers. This study investigates the association between MRI-derived adiposity markers and skeletal muscle volume in relation to breast cancer risk. Using data from 15,669 postmenopausal women in the UK Biobank, the authors report that higher visceral adipose tissue (VAT) and muscle fat infiltration (MFI) are associated with an increased risk of breast cancer.

Comment: The authors should compare their findings with similar studies from other countries to better contextualize the results and highlight population-specific differences or similarities.

Author Response

Comment 1: Obesity is a major global health concern, with strong evidence linking excess body weight to several cancers. This study investigates the association between MRI-derived adiposity markers and skeletal muscle volume in relation to breast cancer risk. Using data from 15,669 postmenopausal women in the UK Biobank, the authors report that higher visceral adipose tissue (VAT) and muscle fat infiltration (MFI) are associated with an increased risk of breast cancer.

The authors should compare their findings with similar studies from other countries to better contextualize the results and highlight population-specific differences or similarities.

Response 1: Yes, we have now revised the Discussion to include direct comparisons with studies from other countries and to highlight population-specific similarities and differences in the associations between adiposity, muscle quality, and breast cancer risk (Page 13, Line 385-391).

Reviewer 3 Report

Comments and Suggestions for Authors

The paper focused on the large, well-characterized population of 15,669 postmenopausal women  and provides great statistical power and high-quality prospective data. However, i do have some concerns here: 

-the authors repeatedly claim this to be the "first" study, while similar large-scale imaging studies exist. Rephrase to avoid overclaiming: "One of the first large-scale studies using MRI-derived composition metrics."

-Limited adjustment for possible residual confounding. Key risk factors have been included, but some important variables have not been controlled for or measured like diet patterns, change in weight over a period of time, statin/metformin use and physical fitness level beyond METs

-The population of the UK Biobank is predominantly White (>90%), healthier, and more affluent than the general population. Discuss more explicitly how this limits applicability to diverse populations.

-Selection bias in MRI subsample, Women who had MRI scans differed significantly from the rest of the cohort in such as race, smoking, HRT use, alcohol intake

-The observational design limits the causal inference, especially since:VAT may be a marker, not a mechanism. MFI could be secondary to other metabolic conditions.

-Spline curves could be interpreted more clearly. The authors claim a "J-shaped" or "exponential" pattern, though this is not necessarily adequately reflected in figures without further explanation. a stronger justification or clearer visual representation of the curve shapes should be provided.

-Results for body composition phenotype are inconsistent like categories of phenotypes were not significantly associated in multivariable models, raising questions regarding
whether the categories are well defined, collinearity between covariates whether phenotypes add value beyond VAT & MFI
-Lack of information about subtypes of breast cancer VAT and MFI may differ by association with ER/PR/HER2 status; however, subtype analyses were not possible.

-Introduction and discussion are excessively long. Parts of it (metabolic mechanisms, MRI theory) are rather long-winded and repetitive.

- Tables & results could be simplified like some tables have a great deal of repeated information, which can get cumbersome for the reader. 

Author Response

Comment #1: The paper focused on the large, well-characterized population of 15,669 postmenopausal women and provides great statistical power and high-quality prospective data. However, i do have some concerns here: 

-the authors repeatedly claim this to be the "first" study, while similar large-scale imaging studies exist. Rephrase to avoid overclaiming: "One of the first large-scale studies using MRI-derived composition metrics."

Response #1: Thank you for the comment. We have made edits.

Comment #2: Limited adjustment for possible residual confounding. Key risk factors have been included, but some important variables have not been controlled for or measured like diet patterns, change in weight over a period of time, statin/metformin use and physical fitness level beyond METs.

Response #2: We agree with the reviewer and have now expanded the Limitations section to explicitly acknowledge these potential unmeasured confounders and their possible influence on our findings (page 14, line 476 to 479). We thank the reviewer for this important suggestion, which has improved the balance and clarity of our discussion.

Comment #3: The population of the UK Biobank is predominantly White (>90%), healthier, and more affluent than the general population. Discuss more explicitly how this limits applicability to diverse populations.

Response #3: This has been addressed as part of the limitations

Comment #4: Selection bias in MRI subsample, Women who had MRI scans differed significantly from the rest of the cohort in such as race, smoking, HRT use, alcohol intake

Response #4: This has been addressed as part of the limitations

Comment #5: The observational design limits the causal inference, especially since: VAT may be a marker, not a mechanism. MFI could be secondary to other metabolic conditions.

Response #5: This has been addressed as part of the limitations

Comment #6: Spline curves could be interpreted more clearly. The authors claim a "J-shaped" or "exponential" pattern, though this is not necessarily adequately reflected in figures without further explanation. a stronger justification or clearer visual representation of the curve shapes should be provided.

Response #6: We thank the reviewer for the constructive comment on the interpretation of the spline curves. We agree that the figures showing the curves alone may not be self-explanatory for the "J-shaped" or "exponential" as described in the text without further clarification.

We have edited the results text to be more explicit in describing the behavior of the curves over the range of the exposure (Page 7, Line 282 – 287).

Comment #7: Results for body composition phenotype are inconsistent like categories of phenotypes were not significantly associated in multivariable models, raising questions regarding
whether the categories are well defined, collinearity between covariates whether phenotypes add value beyond VAT & MFI

Response #7: We appreciate the reviewer’s thoughtful comment. We agree that body-composition phenotypes can vary depending on underlying construct definitions and correlations among adipose and muscle compartments. Our intention in using phenotypes was to model the joint influence of adiposity and muscle health, which cannot be fully captured by VAT or MFI alone.

Importantly, this approach has strong precedent in high-impact body-composition research. For example, Caan et al. (https://pmc.ncbi.nlm.nih.gov/articles/PMC5647152/) and colleagues have consistently demonstrated that combined sarcopenia-adiposity phenotypes provide prognostic information beyond individual compartments or BMI alone. These studies illustrate that interactions between muscle quantity/quality and adiposity are biologically meaningful and clinically relevant, even when main-effect associations appear attenuated.

Comment #8: Lack of information about subtypes of breast cancer VAT and MFI may differ by association with ER/PR/HER2 status; however, subtype analyses were not possible.

Response #8: This has been addressed as part of the limitations

Comment #9: Introduction and discussion are excessively long. Parts of it (metabolic mechanisms, MRI theory) are rather long-winded and repetitive.

Response #9: We appreciate the reviewer’s feedback regarding the length of the Introduction and Discussion. We agree that focus and clarity are essential for readability. At the same time, the conceptual background for this work requires succinct explanation of both the metabolic rationale and the imaging-based characterization of body composition, which together form the foundation for our study hypothesis. Each component currently included directly supports the motivation for evaluating multimodal body-composition phenotypes and their clinical relevance.

Comment #10: Tables & results could be simplified like some tables have a great deal of repeated information, which can get cumbersome for the reader. 

Response #10: We thank the reviewer for this constructive comment. We understand the concern that presenting multiple tables with overlapping variables may feel repetitive. However, we believe the current level of detail is important for transparency and accurate interpretation of our findings.

Round 2

Reviewer 1 Report

Comments and Suggestions for Authors

The Authors have revised the manuscript appropriately and made data available.  

Reviewer 3 Report

Comments and Suggestions for Authors

Thank you for answering my questions. good luck